# Research Progress of Perovskite-Based Bifunctional Oxygen Electrocatalyst in Alkaline Conditions

**DOI:** 10.3390/molecules28207114

**Published:** 2023-10-16

**Authors:** Kailin Fu, Weijian Chen, Feng Jiang, Xia Chen, Jianmin Liu

**Affiliations:** 1Department of Materials Science and Engineering, Jingdezhen Ceramic University, Jingdezhen 333403, China; cwj_yybf@163.com (W.C.); jiangfeng@jci.edu.cn (F.J.); 2Sichuan Volcational College of Cultural Industries, Chengdu 610213, China; chenxia0810@163.com; 3National Engineering Research Center for Domestic & Building Ceramics, Jingdezhen Ceramic University, Jingdezhen 333000, China

**Keywords:** perovskite oxide, bifunctional electrocatalyst, oxygen reduction reaction, oxygen evolution reaction, modification strategy

## Abstract

In light of the depletion of conventional energy sources, it is imperative to conduct research and development on sustainable alternative energy sources. Currently, electrochemical energy storage and conversion technologies such as fuel cells and metal-air batteries rely heavily on precious metal catalysts like Pt/C and IrO_2_, which hinders their sustainable commercial development. Therefore, researchers have devoted significant attention to non-precious metal-based catalysts that exhibit high efficiency, low cost, and environmental friendliness. Among them, perovskite oxides possess low-cost and abundant reserves, as well as flexible oxidation valence states and a multi-defect surface. Due to their advantageous structural characteristics and easily adjustable physicochemical properties, extensive research has been conducted on perovskite-based oxides. However, these materials also exhibit drawbacks such as poor intrinsic activity, limited specific surface area, and relatively low apparent catalytic activity compared to precious metal catalysts. To address these limitations, current research is focused on enhancing the physicochemical properties of perovskite-based oxides. The catalytic activity and stability of perovskite-based oxides in Oxygen Reduction Reaction/Oxygen Evolution Reaction (ORR/OER) can be enhanced using crystallographic structure tuning, cationic regulation, anionic regulation, and nano-processing. Furthermore, extensive research has been conducted on the composite processing of perovskite oxides with other materials, which has demonstrated enhanced catalytic performance. Based on these different ORR/OER modification strategies, the future challenges of perovskite-based bifunctional oxygen electrocatalysts are discussed alongside their development prospects.

## 1. Introduction

With the rapid advancement of science and technology, there has been a notable enhancement in people’s living standards. However, this progress has also resulted in excessive consumption of conventional fossil fuels and low energy conversion efficiency, leading to substantial emissions of carbon dioxide and sulfur dioxide. Consequently, these emissions have given rise to critical issues such as greenhouse effects, acid rain, and global warming. Numerous researchers have dedicated their efforts to investigating electrochemical energy storage and conversion devices, such as fuel cells [1,2,3] and metal-air batteries [4,5,6,7], in response to the exhaustion of resources and environmental deterioration. However, the kinetics of electrochemical systems in the oxygen reduction reaction (ORR) and oxygen evolution reaction (OER) during operation are sluggish, necessitating significant overpotentials to drive their kinetic processes. Until now, the excellent catalytic activities of precious metal (oxide) electrocatalysts, such as platinum-based materials for ORR and Iridium-based and Ruthenium-based materials for OER, are significantly constrained by the scarce resources, high cost, low bifunctional activity, and poor durability [8,9,10]. In order to address these challenges, researchers have devoted considerable efforts to reducing the loading of precious metals in catalysts by employing alloys or carbon supports [11]. However, reducing the loading of precious metals often leads to a simultaneous decline in catalytic activity, posing a significant obstacle for practical applications. At the same time, methodologies for catalyst morphologies, structural characteristics, physical and chemical properties, and screening pathways also need to be considered to design the optimal bifunctional electrocatalysts [12]. Therefore, it is crucial to develop highly efficient and durable bifunctional electrocatalysts that are not dependent on precious metals, addressing the urgent need for sustainable and cost-effective alternatives in the large-scale implementation of renewable energy technologies.

Schuhmann et al. proposed a measure of bifunctional properties, referred to as the bifunctional index (BI), which quantifies the difference in potential required to achieve an OER current density of 10 mA/cm^2^ and an ORR current density of −1 mA/cm^2^ [13,14]. An ideal bifunctional catalyst holds great promise as a crucial component in the next generation of sustainable energy storage devices and shows potential application as a bifunctional catalyst for both ORR and OER [15,16,17,18,19]. In recent years, ABO_3_-based perovskite oxides have attracted significant attention as promising alternatives to high-efficiency bifunctional oxygen electrocatalysts owing to their unique compositional variability, special physicochemical properties, low cost, and environmentally friendliness [20,21,22,23]. Normally, the A-site is occupied by large-radius rare earth or alkali metal ions (such as La, Ca, Sr, Ba), which exhibit higher electronegativity and are 12-fold coordinated with oxygen ions. Meanwhile, the B-site is occupied by transition metal ions (like Co, Fe, Mn, Ni, etc.) located at the octahedral center of a cubic compact stack with a 6-fold oxygen coordination. In particular, partial substitution at the A and/or B sites in perovskite oxides can lead to alterations in the valence states of A- and B-site cations, as well as the formation of oxygen vacancies.

This crystal structure of perovskite oxides offers an opportunity to flexibly modulate their electronic and catalytic properties while exhibiting remarkable stability in alkaline environments [24,25,26,27,28]. However, the electron conductivity of perovskite bifunctional oxygen electrocatalysts at room temperature is insufficient, hindering efficient electron transport that is crucial for ORR/OER reactions and limiting their practical applicability [29,30]. Various strategies have been investigated to address this issue and enhance the catalytic performance and stability of perovskite bifunctional oxygen electrocatalysts, including crystallographic structure tuning, cationic regulation, anionic regulation, nano-processing, and composite processing (Figure 1).

## 2. Crystallographic Structure Tuning

Crystallographic structure tuning has been demonstrated as an effective strategy for enhancing the electrocatalytic performance of perovskite oxides [31,32,33]. Currently, the crystal structure and surface properties of perovskite electrocatalysts can be modified by adjusting the firing temperature of the crystals, thereby effectively improving the catalytic performance in ORR and OER [34,35,36,37]. For instance, tuning the B-O bond property using variation of the annealing temperature, thereby enhancing the catalytic performance of perovskite oxides. Zhou et al. proposed that the rhombohedral structure of LaNiO_3_ perovskite crystal transitions to the cubic structure as the annealing temperature is increased from 400 °C to 600 °C (Figure 2a) [38]. This phase transition directly reflects the influence of annealing temperature on the crystal structure. The length of the O-Ni bond increases with increasing annealing temperature, accompanied by an increase in the Ni-O-Ni bond angle, which results in an improvement in the electrocatalytic activity of LaNiO_3_. Jung et al. eliminated the inhomogeneous spinel surface that originally existed between the amorphous outer layer and the inner cubic phase layer of Ba_0.5_Sr_0.5_Co_x_Fe_1−x_O_3−δ_ (x = 0.2/0.8) by heat-treating at 950 °C in an argon atmosphere (Figure 2b), resulting in a significant improvement that enables the material to achieve the ORR limiting current density of 6.4 mA/cm^2^ at 0.1 V versus RHE, slightly surpassing that of commercial Pt/C catalysts [39].

Tuning the crystallographic structure in perovskite-based oxides is considered an important factor that affects their electrocatalytic performance [40,41,42]. Adjusting the preparation method of perovskite oxides can induce modifications in their crystal structure, morphology, surface properties, and internal defects, thereby influencing the catalytic performance of the electrocatalyst. Currently, several methods have been established for tuning the crystallographic structure of perovskite oxides, including the conventional solid-state method [43], electrospinning technique [44], hydrothermal and solvothermal methods [45], sol-gel method [46], and polymer-assisted method [47]. In addition, Gonell synthesized pure pseudocubic perovskite La_0.7_Sr_0.3_MnO_3_ (pc-LSMO) nanocrystals via the molten salt method, which exhibited high mass-normalized ORR activity of 21.4 A/g _oxide_ at 0.8 V/RHE in alkaline conditions [48]. The author proposed that the improvement in catalytic performance of pc-LSMO can be attributed to its relatively large surface area, high crystallinity, and electron mobility. Kim et al. employed a glycine-nitrate combustion method to synthesize three distinct perovskite oxide crystals, among which the triple perovskite oxide Nd_1.5_Ba_1.5_CoFeMnO_9-d_ (NBCFM) exhibited superior ORR/OER activity characterized by higher current density and lower overpotential owing to its tetragonal crystal structure [49].

## 3. Cationic Regulation

Cationic regulation is a widely employed modification technique for perovskite materials, as reported in previous studies [50,51,52]. The selection of an appropriate introduced cation requires consideration of several factors, such as the cationic radius, the valence state of the cation, and the interaction between the introduced cation and the perovskite parent. Firstly, it is important for the cationic radius to be close to that of the perovskite parent since this contributes to both the integrity and stability of the lattice. An excessively large or small cationic radius has the potential to disrupt the crystal structure of the perovskite parent. Secondly, the valence state of the cation must also match that of the perovskite parent in order to ensure effective replacement and maintain electrical neutrality within the crystal structure. Additionally, accurate control over the ratio of the introduced cation to that of the perovskite parent is imperative to ensure the desired modification effect. However, excessive introduction may result in detrimental consequences such as crystal structure destruction or instability [53,54]. Various approaches have been reported in many research studies, including A-site cation regulation, B-site cation regulation, A-site and B-site cation regulation, and cation vacancies.

### 3.1. A-Site Cation Regulation

The A-site cations of perovskite oxides generally do not directly participate in the oxygen electrode reaction process for ORR/OER; however, they indirectly enhance the catalytic reaction [55,56,57]. The substitution at the A-site has been shown in numerous reports to result in a significant enhancement of catalytic activity [58,59,60]. Partial substitution of A-site cation ions, such as La, Sr, Ca, and Ba, can introduce novel lattice defects that significantly enhance the catalytic activity and stability of perovskite electrocatalysts. Hu et al. [61] demonstrated that the catalytic activity of LaMnO_3_ perovskite oxide-graphene composites could be effectively modulated by incorporating Ca ions without altering the structure of the perovskite. Kumar et al. [62] successfully incorporated Gd ions into the A-site of the double-layer perovskite oxide (GdBa_0.5_Sr_0.5_Co_1.5_Fe_0.5_O_6_), resulting in superior catalytic activity in both ORR and OER, characterized by a high catalytic current density and low overpotential. Cheng et al. [63] found that partial substitution of Sr ions for La in LaCoO_3_ facilitates a preferential arrangement of Co-O-Co bonds, resulting in a transition from a rhombohedral phase to a cubic phase (Figure 3a). Additionally, this substitution leads to an increase in the valence state of Co ions (Figure 3b). The density functional theory calculations indicate that the rearrangement of Co-O-Co bonds and the enhanced valence state of Co ions facilitate an improved overlap between the occupied O 2p valence band and unoccupied Co 3d conduction band, thereby enhancing both Ex situ conductivity of La_0.2_Sr_0.8_CoO_3_ and intrinsic activity for OER.

### 3.2. B-Site Cation Regulation

Numerous studies have demonstrated that the B-site cations in perovskite oxides exhibit direct involvement in the catalytic reactions of ORR and OER [64,65,66,67]. Therefore, partial substitution of B-site cations is a promising strategy for enhancing the catalytic activity of perovskite-based oxide [68,69,70,71]. Ni-doped perovskite oxide La_0.8_Sr_0.2_Mn_1−x_Ni_x_O_3_ (LSMN) was synthesized using the sol-gel method, as reported by Yuan [72] et al. In comparison to undoped La_0.8_Sr_0.2_MnO_3_, LSMN exhibits a higher concentration of oxygen vacancies and demonstrates enhanced bifunctional catalytic activity (Figure 3c,d). In particular, the overall overpotential of LSMN in ORR and OER is significantly low, approaching that of commercial Pt/C.

**Figure 3 molecules-28-07114-f003:**
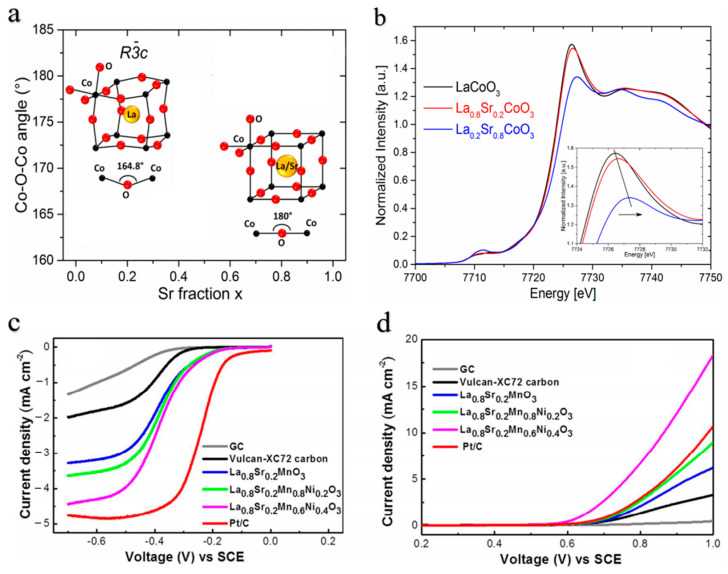
(**a**) Evolution of the pseudocubic (x ≤ 0.4) and cubic (x ≥ 0.6) Co–O–Co angle as a function of the Sr fraction [63]. (**b**) Co-K-edge XANES spectra of the La_1−x_Sr_x_CoO_3_ (x = 0, 0.2, 0.8) at room temperature; the inset shows the main peak region [63]. (**c**) ORR (negative scan) and (**d**) OER (positive scan) polarization profiles of Glassy carbon (GC), VC, La_0.8_Sr_0.2_MnO_3_, La_0.8_Sr_0.2_Mn_0.8_Ni_0.2_O_3_, La_0.8_Sr_0.2_Mn_0.6_Ni_0.4_O_3_, Pt/C at 1600 rpm [72].

Wang et al. utilized the metal cation Ta^5+^ to dope SrCoO_3_ with a cubic perovskite structure, resulting in the formation of SrCo_1−x_Ta_x_O_3−δ_ (x = 0.05/0.1) [73]. This doping strategy effectively compensated for the deficiency of lattice oxygen during high-temperature synthesis and significantly enhanced the phase stability of the material. Duan [74] et al. synthesized a Fe-doped LaCoO_3_ electrocatalyst and optimized the electronic structure of LaCoFeO_3_ catalysts by adjusting the amount of Fe doping. The results demonstrate that the catalytic activity towards OER of LaCo_0.9_Fe_0.1_O_3_ electrocatalyst is significantly enhanced by a doping amount of 0.1 Fe. Furthermore, the authors employed density functional theory simulations and calculations to elucidate the underlying reasons for this improved catalytic activity. It has been discovered that a 10% Fe doping leads to an increase in the strength of covalent bonds between Co 3d and O 2p orbitals, thereby effectively reducing OER overpotential and promoting enhanced OER activity. Sun et al. utilized the metal ions Mn-Ni pairs as dual dopants at B-site cations in lanthanide perovskite-based oxides LaMn_x_Ni_y_Co_z_O_3_ (x + y + z = 1) to enhance the bifunctional ORR and OER activities [75]. The findings demonstrate that the enhanced bifunctional performance of the optimized LaMn_x_Ni_y_Co_z_O_3_ (x:y:z = 1:2:3) catalyst can be attributed to a reduction in the number of e_g_ orbital electron and the modulation of the O 2p-band closer to the Fermi level.

### 3.3. A-Site and B-Site Cation Regulation

The co-regulation of A-site and B-site cations represents a unique approach to enhancing the electrocatalytic performance of perovskite oxide catalysts for ORR and OER. This strategy involves the substitution of cationic components in the A and B sites of the perovskite crystal structure, thereby altering its properties and catalytic behavior [76,77,78]. Li et al. [79] incorporated Bi^2+^ and Fe^3+^ ions into the A and B sites of SrCoO_3_, respectively, exploiting the disparity in ionic radii between Bi^2+^, Fe^3+^, and the original A/B sites. The synthesized Bi_0.15_Sr_0.85_Co_0.8_Fe_0.2_O_3−δ_ powder possesses a surface area of 5.81 m^2^g^−1^ and exhibited excellent ORR and OER activity, low overpotential and favorable chemical kinetics. The specific co-doping strategy induced alterations in surface properties and valence states, leading to enhanced electrocatalytic performance. Shao et al. developed a triple-conducting nanocomposite material, BaCe_0.16_Y_0.04_Fe_0.8_O_3-δ_, using a sol-gel method. This material in cathode exhibited remarkable properties for intermediate-temperature proton ceramic fuel cells (IT-PCFCs). It’s important to note that this composition involved co-doping with Ce and Y at the B-site of BaFeO_3-δ_. This co-doping strategy was employed to stabilize the phase structure and introduce proton conductivity simultaneously. The optimized nanocomposite material in the cathode demonstrated excellent activity and stability for the ORR [80].

Yuan et al. [81] synthesized a bifunctional catalyst, Fe-doped mangan-based perovskite (La_0.8_Sr_0.2_)_0.95_Mn_0.7_Fe_0.3_O_3_ with A-site defect, using the traditional CA-EDTA method. Compared to the pristine La_0.8_Sr_0.2_MnO_3_ catalyst, this modified catalyst demonstrated superior ORR and OER activity under alkaline conditions owing to increased surface oxygen vacancies as well as adjustments in Mn and Fe valence states within the perovskite structure, thereby enhancing catalytic performance. Wei et al. [82] prepared a novel La_1.7_Sr_0.3_Co_0.5_Ni_0.5_O_4+δ_ with a layered perovskite structure that was synthesized using the sol-gel method. In comparison with undoped La_2_NiO_4+δ_, La_1.7_Sr_0.3_Co_0.5_Ni_0.5_O_4+δ_ exhibits significantly enhanced catalytic activity for both ORR and OER, characterized by higher limiting current density, larger positive onset potential, improved half-wave potential, as well as enhanced stability.

### 3.4. Cation Vacancies

The introduction of cation vacancy defects is also regarded as an effective strategy for enhancing the catalytic activity of perovskite oxides [83,84,85,86]. Cation vacancies primarily occur at the A-site in perovskite oxides. By precisely adjusting the cation stoichiometry at the A-site, it is possible to effectively modulate the valence state of ions at the B-site, thereby exposing a greater number of active sites. This adjustment reduces the energy barrier for oxygen migration, enhances oxygen vacancy formation, alters the electronic structure, and ultimately enhances the bifunctional catalytic activity of the perovskite oxides [87,88]. Cheng et al. [89] introduced appropriate A-site cation defects into Sr_x_Co_0.8_Fe_0.2_O_3-δ_ to optimize the occupancy of e_g_ orbitals in B-site transition metals and enhance the generation of oxygen vacancies, thereby improving the bifunctional catalytic performance of perovskite under alkaline conditions, specifically in terms of OER activity.

Zhu [90] et al. introduced A-site cation defects into LaFeO_3_ perovskite oxides to improve their catalytic activity for ORR/OER in alkaline environments. This phenomenon can be attributed to the perturbation of transition metal valence states caused by the introduction of A-site defects. In perovskite-based oxides, A-site defects not only modify the valence states of B-site elements but also influence particle size, surface oxygen vacancies, and the number of active sites available for ORR and OER. Yuan [91] et al. synthesized A-site cation-deficient (La_0.8_Sr_0.2_)_1−x_MnO_3_ (x = 0.02, 0.05) via the sol-gel method, which demonstrated superior bifunctional catalytic activity in alkaline electrolytes compared to La_0.8_Sr_0.2_MnO_3_. The observed outcomes were ascribed to the presence of cation deficiency, which led to a reduction in particle size, an increase in oxygen vacancies, and the attainment of an appropriate Mn valence state. Consequently, these factors facilitate the transport of oxygen ions and improve ORR/OER performance.

## 4. Anionic Regulation

The incorporation of anionic defect engineering, such as oxygen vacancies or anion doping, has been previously demonstrated to positively influence the bifunctional electrocatalytic activity and stability towards ORR/OER in in perovskite oxides [92,93,94,95,96]. The presence of anionic defects can enhance the capability of electron trapping, fine-tune energy band structures, and optimize catalytic reaction pathways in perovskite oxides. Oxygen vacancies are intricately linked to the electronic structure and valence state of B-site transition metals within the perovskite framework, with a substantial abundance of oxygen vacancies serving as active sites for adsorption and reaction processes. These introduced vacancies interact with adsorbed intermediates during catalytic reactions, facilitating efficient charge transfer. Chen et al. synthesized a new type of oxygen-deficient BaTiO_3−x_ perovskite electrocatalyst with a unique hexahedral crystal structure via the sol-gel method followed by reductive heat treatment at 1300 °C in vacuum [97]. They proposed that the presence of oxygen vacancies in h-BaTiO_3−x_ crystal structures facilitates the adsorption of reactants and enhances charge transfer. The results indicated that the oxygen-deficient BaTiO_2.76_ electrocatalyst exhibited excellent bifunctional catalytic activity, particularly surpassing the performance of the IrO_2_ catalyst in terms of OER activity at relatively low potential (<1.6 V).

Heat treatment under moderately reducing conditions enables the controlled incorporation of varying concentrations of oxygen vacancies in perovskite oxides while preserving their original crystal structure. Wang [98] et al. successfully transformed Pr_0.5_Ba_0.5_MnO_3−δ_ (PBM), a perovskite oxide, into layered PrBaMn_2_O_5+δ_ (H-PBM) by annealing PBM in H_2_ at 800 °C for 15 h. This method allowed for the generation of a high concentration of oxygen vacancies in the layered H-PBM material without altering its chemical composition. Remarkably, H-PBM exhibited exceptional catalytic activity towards both ORR and OER, as evidenced by significant enhancements in onset potential and catalytic limiting current density (Figure 4b,c).

A single phase of Ca_2_Mn_2_O_5_ was synthesized by Kim [99] et al. at 350 °C and 5% H_2_/Ar, resulting in the formation of oxygen-deficient perovskite oxide Ca_2_Mn_2_O_5_ with intrinsic molecular-level pores at the oxygen-deficient sites (Figure 4d), facilitating favorable ion transport for OER [100,101,102].

Introducing non-metallic ions such as N, P, and S is an effective method to optimize the electron configuration, alter the valence state of ions, and enhance the activity of perovskite electrocatalysts [103,104,105,106]. Peng et al. [107] prepared sulfur-doped perovskite CaMnO_3_ (CMO/S) nanotubes using an electrospinning technique followed by calcination and sulfurization treatment (Figure 5a). These catalysts exhibit enhanced bifunctional oxygen electrocatalytic activity and stability in alkaline solutions compared to the pristine CaMnO_3_ (Figure 5b,c). The authors proposed that these results were attributed to sulfur doping, which can substitute oxygen atoms to enhance intrinsic electrical conductivity and introduce abundant oxygen vacancies to provide adequate catalytically active sites. The sulfur-doped perovskite oxides LaCoO_3_ (S-LCO), synthesized by Ran et al., exhibited significantly enhanced bifunctional oxygen electrocatalytic activities, which can be attributed to the introduction of S-dopants and oxygen defects [108]. First-principles calculations and experiments were employed to demonstrate that the S-dopant enhances the lattice distortion while simultaneously regulating the electronic filling of Co^3+^ e_g_ states.

## 5. Nano-Processing

Nano-processing has been demonstrated as an effective strategy for enhancing the bifunctional catalytic activity of perovskite-based oxide electrocatalysts, owing to their high specific surface area and well-defined pore structure. These characteristics facilitate efficient electron and oxygen transfer processes while also providing stability to specific oxygen-containing intermediates [109,110,111,112]. However, the conventional synthesis methods for preparing perovskite oxides, such as solid-phase synthesis and sol-gel method, often suffer from drawbacks, including large particle size and unclear morphological characteristics. The challenge lies in the precise design of particle size, morphology, composition, surface properties, and porosity of nanostructured materials to achieve nanoporous perovskite electrocatalysts.

At present, the main synthetic approaches employed for the preparation of perovskite nanomaterials include soft template [113], hard template [114], colloid crystal template [115], electrospinning [116], and hydrothermal methods [117]. These approaches effectively yield perovskite nanomaterials with reduced particle size, thereby increasing their surface area and exposing a greater number of active sites, consequently enhancing their bifunctional catalytic activity [118].

The hollow structure of porous perovskite oxide nanotubes La_0.75_Sr_0.25_MnO_3_ (PNT-LSM) was fabricated by Xu et al. using the electrospinning technique, followed by calcination at 650 °C for 3 h (Figure 6a–c) [119]. This unique structure of the perovskite electrocatalyst facilitates an increased specific surface area and exposes a greater number of active sites, thereby enhancing bifunctional catalytic activity. Liu et al. [120] have developed a mesoporous/macroporous nanotube-structured perovskite oxide, La_0.5_Sr_0.5_CoO_3−x_ (HPN-LSC) via electrospinning, serving as a bifunctional electrocatalyst for both ORR and OER in Li-oxygen batteries. This electrocatalyst structure enhances both electron conduction and the specific surface area, providing more space for product precipitation or storage (Figure 6d,e).

The nanoscale perovskite oxide La_0.6_Sr_0.4_Co_0.2_Fe_0.8_O_3_ (LSCF) was synthesized via the sol-gel method by Cheng et al. [121], and its potential as a cathode catalyst for non-aqueous lithium-air batteries was investigated. The results indicate that the LSCF nanoparticles, with an average particle size of 60 nm, are beneficial for both ORR and OER processes, resulting in a significant reduction in charge–discharge overpotential. Mesoporous nanofibers (PBSCF-NF) were successfully synthesized by Bu et al. [122] and exhibited excellent electrochemical performance in a 6 mol/L KOH electrolyte. At a current density of 300 mA/cm^2^, PBSCF-NF demonstrated a higher power density of 127 mW/cm^2^ compared to the pristine air electrode (109 mW/cm^2^). This enhancement can be attributed to the increased number of catalytic sites and enhanced reactant-catalyst contact facilitated by the nanostructures. Jung et al. synthesized La_x_(Ba_0.5_Sr_0.5_)_1−x_Co_0.8_Fe_0.2_O_3-δ_ (BSCF) nanoparticles with a size of approximately 50 nm by precisely controlling the lanthanum concentration and calcination temperature to manipulate oxide defect chemistry and particle growth mechanisms [123]. The resulting electrocatalyst demonstrated a more than twenty-fold increase in gravimetric activity (A/g) compared to IrO_2_ during half-cell testing using 0.1 M KOH electrolyte while also surpassing the charge/discharge performance of Pt/C (20 wt%) in zinc-air full-cell testing employing 6 M KOH electrolyte.

## 6. Composite Processing

In general, composite electrocatalysts derived from the combination of perovskite oxides with carbon materials, transition metals, noble metals, etc., exhibit significantly enhanced catalytic activity compared to pristine perovskite electrocatalysts [124,125,126]. This composite processing facilitates the modification of both the electronic structure and active sites of catalysts, thereby serving as an effective strategy for enhancing the electrocatalytic activity of perovskite oxides.

### 6.1. Dual Components Integrated Electrocatalyst

To tackle the significant challenge of poor electrical conductivity in perovskite oxides, the researchers propose a strategic approach for fabricating perovskite/carbon-based dual components integrated electrocatalysts using composite processing [127,128,129,130,131]. The bifunctional activity of perovskite/carbon-based dual components integrated electrocatalysts has been consistently demonstrated to surpass that of single-component perovskite or carbon materials in numerous reports. For instance, Park et al. have developed a bifunctional composite electrocatalyst composed of porous perovskite oxide La_0.5_Sr_0.5_Co_0.8_Fe_0.2_O_3_ (LSCF-PR) nanorods and nitrogen-doped reduced graphene oxide (NRGO) [132]. In this study, LSCF-PR is embedded between NRGO sheets to form an efficient composite morphology of LSCF-PR/NRGO, which exhibits excellent catalytic performance for both ORR and OER in alkaline media.

In addition, the utilization of in situ grown perovskite/carbon-based dual components integrated electrocatalyst represents an effective strategy to further enhance catalytic performance. Following this concept, Wu et al. first synthesized cobalt-free oxide SrFe_0.85_Ni_0.05_Ti_0.1_O_3-δ_ using chemical vapor deposition, and subsequently, modified carbon nanotubes were formed in situ on its surface [133]. The resulting perovskite/carbon-based dual components integrated electrocatalyst exhibit superior catalytic activity in alkaline media compared to single-component parent perovskite SrFe_0.85_Ni_0.05_Ti_0.1_O_3−δ_ or carbon nanotubes alone, due to the potential synergy between the perovskite/carbon nanotube phases that facilitate rapid charge transfer rates, increased surface areas, and exposed active sites. Furthermore, numerous perovskite/carbon-based dual components integrated electrocatalysts have been extensively documented in the literature. For example, perovskite La(Co_0.55_Mn_0.45_)_0.99_O_3−δ_ nanorods integrated with N-doped reduced graphene oxide composites [134] and Ba_0.5_Sr_0.5_Co_0.8_Fe_0.2_O_3−δ_ (BSCF) combined with g-C_3_N_4_-Vulcan Carbon composites [135] have also been investigated.

To date, a range of carbon-supported perovskite oxides have been synthesized, exhibiting enhanced ORR and OER activity compared to pure perovskite oxides or carbon alone. Numerous studies attribute this phenomenon to the synergistic effect resulting from the interfacial interaction between these two components, leading to superior performance than that of each individual component [136,137]. To elucidate the synergistic effect of perovskite/carbon-based dual components integrated electrocatalyst, Shao et al. [138] classified it into three aspects: the ligand effect (Figure 7a), the formation of interfacial heterostructure (Figure 7b) and the spillover effect (Figure 7c). Ligand effects pertain to the modification of electronic structure by ligand molecules in a material, thereby facilitating electron transfer. For instance, Fabbri’s group observed this phenomenon in the BSCF perovskite/acetylene black (AB) carbon composite, where AB was found to modulate the Co oxidation state of BSCF, reduce it, enhance its inherent conductivity, and improve its oxide adsorption capacity [139]. Consequently, the composite catalyst exhibited heightened ORR/OER activity. The formation of interfacial heterostructures, such as covalent bonds or new phases at the interface, serves as the underlying mechanism for synergistic improvements in both ORR and OER performance. The well-dispersed perovskite-based LaMnO_3_ nanoparticles were synthesized by Liu et al. via physical mixing of carbon and LaMnO_3_ nanoparticles, followed by loading onto modified carbon black and subsequent sintering at various temperatures [140]. Among these catalysts, the carbon-LaMnO_3_ hybrid with a mass ratio of 2:3 exhibited superior electrocatalytic activity for ORR. This remarkable enhancement in ORR electrocatalytic activity can be attributed to the formation of C-O m (M = La, Mn) covalent bonds between metal oxide nanoparticles and the carbon support, which effectively enhances ORR kinetics. The spillover effect involves the diffusion and transport of reactants at the perovskite/carbon interface. Perovskite materials adsorb and catalyze reactants while carbon materials facilitate their diffusion to a wider area, increasing coverage on catalyst surfaces and enhancing reaction rates. These aforementioned effects synergistically enhance the overall catalytic activity of the composite catalyst, thereby significantly improving its efficiency in facilitating both ORR and OER [138].

### 6.2. Multiple Components Integrated Electrocatalyst

Multiple components integrated electrocatalysts consist of at least three components, wherein the electrocatalytic activity for ORR and OER originates from distinct active sites. The composite nature of these catalysts provides opportunities to independently design and construct specific ORR and OER active sites, thereby benefiting from a wider range of available active sites, more versatile regulation strategies, and potentially additional assistance in fine-tuning the ORR/OER activities. Consequently, composite electrocatalysts generally exhibit superior performances compared to their single-component counterparts. In addition to interacting with carbon-based materials, perovskites can also be combined with transition metal compounds or precious metals to load onto carbon-based materials, thereby enhancing the catalytic activity of composites for ORR/OER.

Perovskite/carbon/transition metal integrated electrocatalysts have attracted significant attention as bifunctional catalysts for ORR and OER [141,142,143,144,145]. Hua et al. reported that a three components-integrated catalyst consisting of La_0.45_Sr_0.45_Mn_0.9_Fe_0.1_O_3−δ_ perovskite and Fe_3_C and Carbon exhibited excellent catalytic activity for both ORR and OER under alkaline conditions [146]. A bifunctional electrocatalyst composed of perovskite oxides LaCoO_3_ and transition metal oxide MnO_2_ and Vulcan XC-72 was developed by Benhangi et al. in saturated 6 M KOH, the synergistic effect between these materials is evident as the bifunctional activity of MnO_2_/LaCoO_3_/Vulcan XC-72 surpasses that of MnO_2_ or LaCoO_3_ alone [147].

Furthermore, the researchers synthesized composite materials of perovskite oxides with precious metals such as Pt/C or IrO_2_. Platinum is widely recognized as an exceptional electrocatalyst for ORR, despite its drawbacks, including high cost and limited durability. By integrating perovskite oxides with precious metals to load onto carbon-based materials, it becomes possible to design a bifunctional electrocatalyst that exhibits superior performance [148,149,150]. The potential of Perovskite/carbon/precious metals integrated electrocatalyst has been demonstrated in relevant literature [151,152]. The precious metal Pt and Sr(Co_0.8_Fe_0.2_)_0.95_P_0.05_O_3−δ_ (SCFP) and C-12 were physically mixed via ball milling by Wang et al. [153], resulting in the synthesis of a bifunctional ORR/OER electrocatalyst with exceptional performance. This apparent enhancement in catalytic activity can be attributed to accelerated electron transfer, an abundance of surface oxygen vacancies, increased active sites, and reduced energy barriers due to the spillover effect between Pt and SCFP. Zhu [154] et al. employed a facile ultrasonic mixing technique to combine the Pt/C catalyst with the perovskite oxide Ba_0.5_Sr_0.5_Co_0.8_Fe_0.2_O_3−δ_ (BSCF) at an optimal ratio, resulting in a composite that exhibits excellent ORR activity. This integration of Pt/C and BSCF enhances the electron transfer mechanism of BSCF and facilitates synergistic catalysis between the two catalysts, thereby significantly boosting the bifunctional oxygen catalytic performance of BSCF.

In addition to the aforementioned methods of compounding perovskite materials, it is also feasible to effectively compound perovskite materials with other substances such as metal-organic frameworks (MOF), perovskites [155], and spinels [156], thereby enhancing their bifunctional oxygen catalytic activity.

The potentials of various perovskite-based bifunctional electrocatalysts with different modification strategies are presented in Table 1, at a current density of 10 mA/cm^2^ for ORR and −1 mA/cm^2^ for OER, along with the corresponding potential difference (ΔE). In a way, a smaller potential difference (ΔE) indicates superior bifunctional catalytic activity. These data revealed that the activity of perovskite-based bifunctional electrocatalysts can be significantly influenced by cationic regulation, composite processing, and nano-processing. Specifically, the regulation of B-site cations tends to have a more pronounced effect on the catalytic activity of perovskite bifunctional electrocatalysts. Taking the LaMn_x_Ni_y_Co_z_O_3_ (x:y:z = 1:2:3) electrocatalyst as an example, its potential difference (ΔE) was calculated as 0.76 V (1.60–0.84 V), which is lower than the other samples listed in Table 1. This remarkable performance is attributed to the dual Mn-Ni dopant at the B-site, which allows precise tuning of the catalyst’s electronic structure. Consequently, this electrocatalyst not only exhibits outstanding activity for both ORR and OER but also serves as a model for further investigations into structure–activity relationships.

## 7. Conclusions and Perspectives

In this paper, we provided an overview of the recent progress in bifunctional perovskite-based electrocatalysts for both oxygen reduction and oxygen evolution reactions under alkaline conditions. We present and analyze various modification strategies for manipulating perovskite-based oxides to enhance their bifunctional catalytic activity and stability, including crystallographic structure tuning, cationic regulation, anionic regulation, nano-processing, and composite processing. Among these strategies, the regulation of B-site cations has demonstrated the most significant influence on enhancing the catalytic activity of perovskite-based bifunctional electrocatalysts. Further, cationic regulation, composite processing, and nano-processing have also exhibited a greater impact on improving their catalytic activity. Despite this, related studies are still in the experimental stage, and there are still numerous challenges that remain for large-scale commercial applications. The practical applications often necessitate the comprehensive utilization of multiple modification strategies to attain superior performance owing to the multifactorial nature that influences the catalytic activity of perovskite-based oxides. Therefore, the integration of these strategies in the development of perovskite-based oxides has the potential to significantly enhance their electrocatalytic performance. Furthermore, it is imperative to conduct additional experimental investigations and theoretical calculations in order to comprehensively explore the bifunctional catalytic mechanism of perovskite oxides and to design more efficient nanostructured perovskite-based electrocatalysts. With the advancement of research, widespread applications of bifunctional perovskite-based electrocatalysts can be anticipated in the foreseeable future.

## Figures and Tables

**Figure 1 molecules-28-07114-f001:**
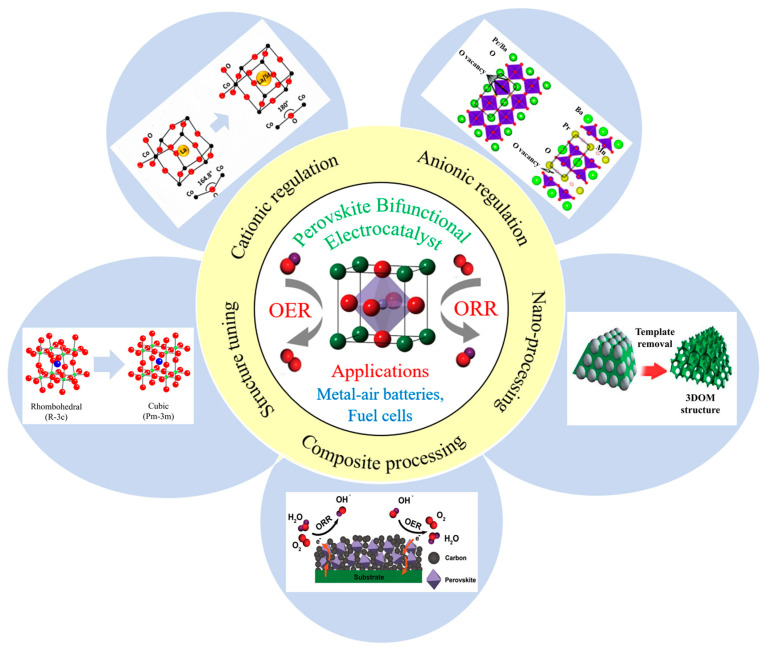
Modification strategy of perovskite bifunctional electrocatalyst for ORR/OER.

**Figure 2 molecules-28-07114-f002:**
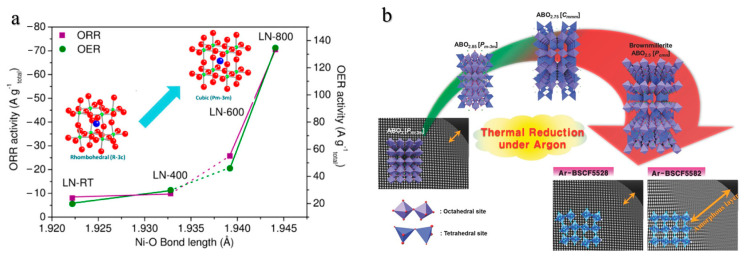
(**a**) Correlation of the ORR/OER activity of LaNiO_3−δ_ perovskites with their crystallographic structure [38]. (**b**) Schematic diagrams of the surface structural changes according to heat treatment at 950 °C for 24 h in argon atmosphere for Ba_0.5_Sr_0.5_Co_0.8_Fe_0.2_O_3−δ_ and Ba_0.5_Sr_0.5_Co_0.2_Fe_0.8_O_3−δ_ [39].

**Figure 4 molecules-28-07114-f004:**
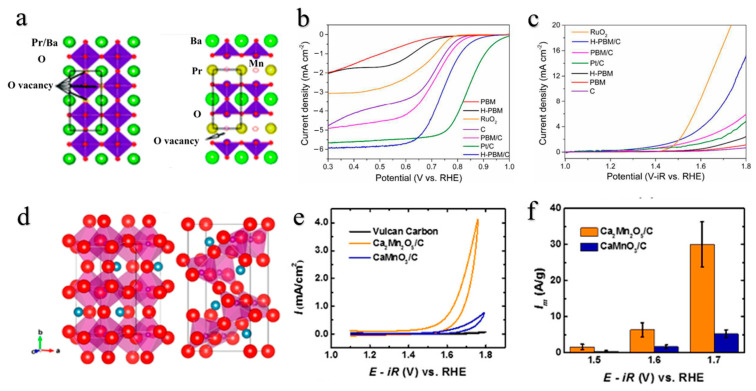
(**a**) Unit cell structures of Pr_0.5_Ba_0.5_MnO_3−δ_ (PBM, left) and PrBaMn_2_O_5+δ_ (H-PBM, right); Oxygen vacancies are represented by arrows. (**b**) ORR and (**c**) OER polarization curves of PBM, H-PBM, RuO_2_, C, PBM/C, Pt/C, and H-PBM/C [98]. (**d**) Unit cell structures of CaMnO_3_ (left) and Ca_2_Mn_2_O_5_ (right); (**e**) OER performance of Ca_2_Mn_2_O_5_/C, CaMnO_3_/C, and Vulcan carbon XC-72; (**f**) mass activities at various applied potentials [99].

**Figure 5 molecules-28-07114-f005:**
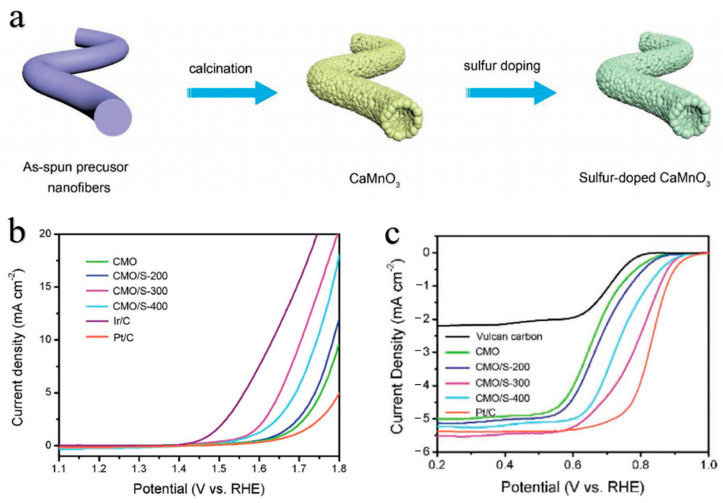
(**a**) Scheme of the formation of CMO/S. (**b**) OER polarization curves of CMO, CMO/S-200, CMO/S-300, CMO/S-400, Pt/C, and Ir/C measured in N_2_-saturated 0.1 M KOH solution at a scan rate of 5.0 mV s^−1^. (**c**) ORR polarization curves of Vulcan XC-72, CMO, CMO/S-200, CMO/S-300, CMO/S-400, and Pt/C in O_2_-saturated 0.1 M KOH solution at a scan rate of 5 mV s^−1^ and rotation rate of 1600 rpm [107].

**Figure 6 molecules-28-07114-f006:**
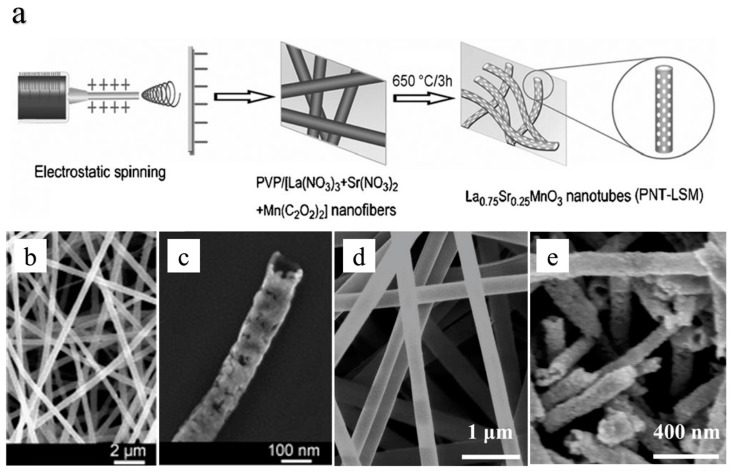
(**a**) the synthesis strategy of the PNT-LSM catalyst [119]. (**b**) As-electrospun composite fibers and (**c**) PNT-LSM after calcination at 650 °C for 3 h [119]. (**d**) As-obtained electrospun composite nanofibers and (**e**) HPN-LSC after thermal treatment at 750 °C for 3 h in air [120].

**Figure 7 molecules-28-07114-f007:**
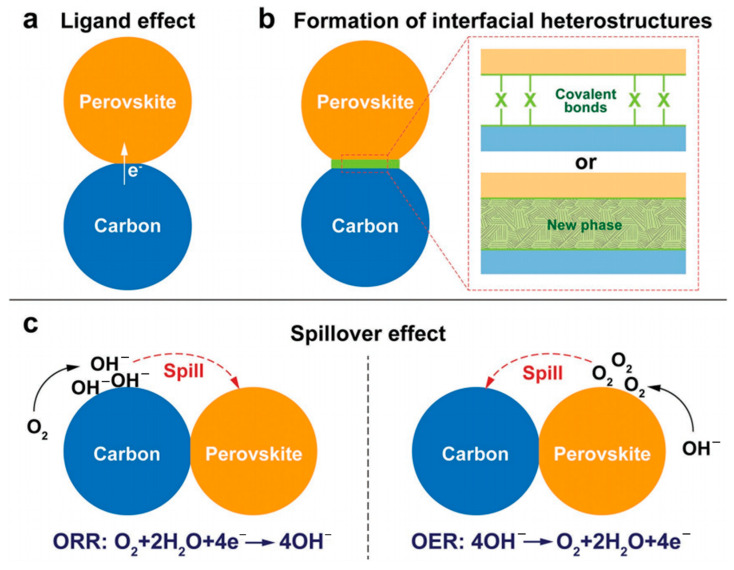
Schematic illustration of possible origins of the synergistic effect between the perovskite and carbon. (**a**) Ligand effect. (**b**) Formation of interfacial heterostructures. (**c**) Spillover effect [138].

**Table 1 molecules-28-07114-t001:** Assessment of perovskite-based bifunctional catalysts in 0.1 M KOH for the above reported.

Catalyst	Modification Strategy	ORR Potential (V) @ −1 mA cm^−2^	OER Potential (V) @ 10 mA cm^−2^	ΔE (V)	Reference
LaNiO_3−δ_	Crystallographic Structure Tuning	−0.25 vs. Ag/AgCl	0.73 vs. Ag/AgCl	0.98	[38]
La_0.8_Sr_0.2_Co_0.4_Mn_0.6_O_3_	B-site regulation	0.81 vs. RHE	1.72 vs. RHE	0.91	[52]
La_0.75_Sr_0.25_Mn_0.5_Fe_0.5_O_3_	nano-processing and B-site regulation	0.74 vs. RHE	1.66 vs. RHE	0.92	[69]
LaMn_x_Ni_y_Co_z_O_3_ (x:y:z = 1:2:3)	B-site regulation	0.84 vs. RHE	1.60 vs. RHE	0.76	[75]
(La_0.8_Sr_0.2_)_0.95_Mn_0.5_Fe_0.5_O_3_	A-site deficiency and B-site regulation	0.12 vs. Ag/AgCl	0.89 vs. Ag/AgCl	0.77	[91]
Vacancy-induced LaMnO_3_	anionic regulation	0.94 vs. RHE	1.84 vs. RHE	0.90	[96]
nsLaNiO_3_/NC	nano-processing	0.74 vs. RHE	1.62 vs. RHE	0.88	[110]
Ni_3_S_2_/PrBa_0.5_Sr_0.5_Co_2_O_5+δ_	composite processing	0.81 vs. RHE	1.63 vs. RHE	0.82	[124]
La(Co_0.55_Mn_0.45_)_0.99_O_3−δ_/NrGO	composite processing	0.84 vs. RHE	1.72 vs. RHE	0.88	[134]
10%g-C_3_N_4_-LaNiO_3_	composite processing	-0.32 vs. SCE	0.76 vs. SCE	1.08	[136]

## Data Availability

Not applicable.

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
