# Peer review of "Research Progress of Perovskite-Based Bifunctional Oxygen Electrocatalyst in Alkaline Conditions"

_molecules, 2023, doi:10.3390/molecules28207114_

Round 1
Reviewer 1 Report
The manuscript of K. Fu et al. discusses the different strategies of the improvement of the performance of the perovskite based electrocatalysts. The perovskite are important materials with applications in different practically relevant processes. Especially, they are promising as electrocatalytic for OER and OER processes This is a rather tutorial review, and its advantage is showing the different strategies the possibilities tuning the catalytic performance and stability of perovsckite electrocatalysts thorough crystallographic structure tuning, cationic regulation, anionic regulation, nano-processing and composite fabrication with other type materials. In particular, different kinds of components for the composite perovskite preparation have been described. This manuscript may be published in Molecules after minor revision by addressing the following issues.
1. Please provide the justification of position of review among other relevant review.
2. Most references in this work have been published in the period 2012-2019. Please add the more modern literature with an appropriate citation in the review.
3. Please provide some comparative evaluation of different strategies for an enhancement of performance of the bifunctional perovskite electrocatalysts. For this, including a few tables for a better illustration of the assessment of the best method will be useful.
English language is quite readable in this manuscript.
Author Response
Reviewer 1#
Comments and Suggestions for Authors
The manuscript of K. Fu et al. discusses the different strategies of the improvement of the performance of the perovskite based electrocatalysts. The perovskite are important materials with applications in different practically relevant processes. Especially, they are promising as electrocatalytic for OER and OER processes This is a rather tutorial review, and its advantage is showing the different strategies the possibilities tuning the catalytic performance and stability of perovsckite electrocatalysts thorough crystallographic structure tuning, cationic regulation, anionic regulation, nano-processing and composite fabrication with other type materials. In particular, different kinds of components for the composite perovskite preparation have been described. This manuscript may be published in Molecules after minor revision by addressing the following issues.
Comments:
- Please provide the justification of position of review among other relevant review.
Response: Thank you for your comment. More attentions have been paid to synthesis methods, types and application of perovskite-based bifunctional electrocatalyst in other relevant review. However, there is a scarcity of reviews that delve into the modification strategies employed for perovskite-based bifunctional electrocatalysts in order to enhance their performance in both oxygen evolution reaction (OER) and oxygen reduction reaction (ORR) under alkaline conditions. In this manuscript, we focus on the diverse modification strategies that aim at enhancing the catalytic activity of perovskite-based bifunctional electrocatalysts in alkaline conditions. These strategies including crystallographic structure tuning, cationic regulation, anionic regulation, nano-processing and composite processing. This manuscript provides readers with additional research ideas and unique perspectives. And this is different from the preparation methods that have been reported in other literature, such as the solid-state method and sol-gel method. Especially, the direct effects of modification strategies on potential difference (ΔE=EOER–EORR) between EORR (-1 mA/cm²) and EOER (10 mA/cm²) performance are reviewed, which is the difference between this review and others.
Among these strategies, the regulation of B-site cations has demonstrated the most significant influence on enhancing the catalytic activity of perovskite-based bifunctional electrocatalysts. Besides, cationic regulation, composite processing, and nano-processing also have exhibited a greater impact on improving their catalytic activity. (Line 485, highlight in yellow.)
- Most references in this work have been published in the period 2012-2019. Please add the more modern literature with an appropriate citation in the review.
Response: We sincerely appreciate the valuable comments. We have checked the references carefully and cited some modern literatures (Ref. 7, Ref. 12, Ref. 19, Ref. 80, Ref. 86, Ref. 96 and Ref. 156) in the revised manuscript.
(1) Ref. 7 (Zhu, Z.; Song, Q.; Xia, B.; Jiang, L.; Duan, J.; Chen, S., Perovskite Catalysts for Oxygen Evolution and Reduction Reactions in Zinc-Air Batteries. Catalysts 2022, 12, (12), 1490. DOI:10.3390/catal12121490), highlight in yellow, Line 42.
(2) Ref. 12 (Guan, D.; Wang, B.; Zhang, J.; Shi, R.; Jiao, K.; Li, L.; Wang, Y.; Xie, B.; Zhang, Q.; Yu, J.; Zhu, Y.; Shao, Z.; Ni, M., Hydrogen society: from present to future. Energy Environ. Sci. 2023. DOI: 10.1039/d3ee02695g), highlight in yellow, Line 55.
(3) Ref. 19 (Chen, T. W.; Kalimuthu, P.; Anushya, G.; Chen, S. M.; Ramachandran, R.; Mariyappan, V.; Muthumala, D. C., High-Efficiency of Bi-Functional-Based Perovskite Nanocomposite for Oxygen Evolution and Oxygen Reduction Reaction: An Overview. Materials 2021, 14, (11), 2976. DOI:10.3390/ma14112976.), highlight in yellow, Line 64.
(4) Ref. 80 (Zou, D.; Yi, Y.; Song, Y.; Guan, D.; Xu, M.; Ran, R.; Wang, W.; Zhou, W.; Shao, Z., The BaCe0.16Y0.04Fe0.8O3−δ nanocomposite: a new high-performance cobalt-free triple-conducting cathode for protonic ceramic fuel cells operating at reduced temperatures. J. Mater. Chem. A 2022, 10, (10), 5381-5390. DOI: 10.1039/d1ta10652j), highlight in yellow, Line 210.
(5) Ref. 86 (Wang, H.; Chen, X.; Huang, D.; Zhou, M.; Ding, D.; Luo, H., Cation Deficiency Tuning of LaCoO3 Perovskite as Bifunctional Oxygen Electrocatalyst. ChemCatChem 2020, 12, (10), 2768-2775. DOI: 10.1002/cctc.201902392.), highlight in yellow, Line 224.
(6) Ref 96 (Devi, V. S.; Athika, M.; Elumalai, P., Vacancy‐induced LaMnO3 Perovskite as Bifunctional Air‐breathing Electrode for Rechargeable Lithium‐Air Battery. ChemistrySelect 2022, 7, (33), e202202554. DOI: 10.1002/slct.202202554.), highlight in yellow, Line 249.
(7) Ref. 156 (Kubba, D.; Ahmed, I.; Kour, P.; Biswas, R.; Kaur, H.; Yadav, K.; Haldar, K. K., LaCoO3 Perovskite Nanoparticles Embedded in NiCo2O4 Nanoflowers as Electrocatalysts for Oxygen Evolution. ACS Appl. Nano Mater. 2022, 5, (11), 16344-16353. DOI: 10.1021/acsanm.2c03395.), highlight in yellow, Line 461.
- Please provide some comparative evaluation of different strategies for an enhancement of performance of the bifunctional perovskite electrocatalysts. For this, including a few tables for a better illustration of the assessment of the best method will be useful.
Response: This is a very good suggestion. We have summarized the catalytic performances of various perovskite-based bifunctional electrocatalysts with different modification strategies and listed a Table in the revised manuscript. (Line 465, Table 1, highlight in yellow)
The potentials of various perovskite-based bifunctional electrocatalysts with different modification strategies are presented in Table 1, at a current density of 10 mA/cm² for ORR and -1 mA/cm² for OER, along with the corresponding potential difference (ΔE). In a way, a smaller potential difference (ΔE) indicates superior bifunctional catalytic activity. These data revealed that the activity of perovskite-based bifunctional electrocatalysts can be significantly influenced by cationic regulation, composite processing, and nano-processing. Specifically, the regulation of B-site cations tends to have a more pronounced effect on the catalytic activity of perovskite bifunctional electrocatalysts. Taking the LaMnxNiyCozO3 (x:y:z=1:2:3) electrocatalyst as an example, its potential difference (ΔE) was calculated as 0.76 V (1.60 V–0.84 V), which is lower than the other samples listed in the Table 1. This remarkable performance is attributed to the dual Mn-Ni dopant at the B-site, which allows precise tuning of the catalyst's electronic structure. Consequently, this electrocatalyst not only exhibits outstanding activity for both ORR and OER but also serves as a model for further investigations into structure–activity relationships.
Reviewer 2 Report
In this review, the authors summarize the recent progresses of perovskite oxides for bifunctional oxygen electrocatalyst in alkaline conditions. The strategies of crystal structure tuning, cationic regulation, anionic regulation, and nano-processing are demonstrated. However, some important issues must be solved for the improvement of this manuscript.
1. The words in Figure 1 are too small and some words should be removed such as “c”. Also, the quality of other pictures should be improved.
2. Besides the discussed strategies, some other methodologies should also be included and demonstrated, such as macroscopic physiochemical properties, molecular-level structures, electronic structures and screening pathways. The authors should refer to this perspective (DOI: 10.1039/d3ee02695g) and the papers cited in this perspective.
3. For the application of OER and ORR in high-temperature SOFC/SOEC/PCFC/PCEC, more texts should be discussed since this application is very important. The authors can refer to this work (10.1039/D1TA10652J).
Minor editing of English language required
Author Response
Reviewer 2#
Comments and Suggestions for Authors
In this review, the authors summarize the recent progresses of perovskite oxides for bifunctional oxygen electrocatalyst in alkaline conditions. The strategies of crystal structure tuning, cationic regulation, anionic regulation, and nano-processing are demonstrated. However, some important issues must be solved for the improvement of this manuscript.
Comments:
- The words in Figure 1 are too small and some words should be removed such as “c”. Also, the quality of other pictures should be improved.”
Response: Thank you for your comment. The typo of words such as “c” is removed in this revised manuscript. In addition, we have improved the resolution of both Figure 1 and Figure 2.
- Besides the discussed strategies, some other methodologies should also be included and demonstrated, such as macroscopic physiochemical properties, molecular-level structures, electronic structures and screening pathways. The authors should refer to this perspective (DOI: 10.1039/d3ee02695g) and the papers cited in this perspective.
Response: We sincerely appreciate the valuable comments. A discussion of this literature is also added in the revised manuscript. At the same time, methodologies for catalyst morphologies, structural characteristics, physical and chemical properties, and screening pathways also need to be considered to design the optimal bifunctional electrocatalysts. The reference (DOI: 10.1039/d3ee02695g) has been cited in the revised manuscript. (Line 52-55, highlight in yellow.)
- For the application of OER and ORR in high-temperature SOFC/SOEC/PCFC/PCEC, more texts should be discussed since this application is very important. The authors can refer to this work (10.1039/D1TA10652J).”
Response: Thank you for your suggestion. The reference (DOI: 10.1039/d1ta10652j) has been cited in this revised manuscript. (Line 209, highlight in yellow.) A discussion of this literature is also added in the revised manuscript. (Line 203, highlight in yellow.)
Shao et al. developed a triple-conducting nanocomposite material, BaCe0.16Y0.04Fe0.8O3-δ, using a sol-gel method. This material in cathode exhibited remarkable properties for intermediate-temperature proton ceramic fuel cells (IT-PCFCs). It's important to note that this composition involved co-doping with Ce and Y at the B-site of BaFeO3-δ. This co-doping strategy was employed to stabilize the phase structure and introduce proton conductivity simultaneously. The optimized nanocomposite material in cathode demonstrated excellent activity and stability for the ORR.